# Occurrence, Biological Characteristics, and Annual Dynamics of *Atherigona orientalis* (Schiner 1968) (Diptera: Muscidae) in China

**DOI:** 10.3390/insects16090931

**Published:** 2025-09-04

**Authors:** Zihao Zhou, Yujie Luo, Jiawei Qin, Xintong Wang, Shuaijun Ning, Jing He, Qiong Zhou

**Affiliations:** College of Life Sciences, Hunan Normal University, Changsha 410006, China; zihao_zhou@aliyun.com (Z.Z.); 19313093132@163.com (Y.L.); akita_q@163.com (J.Q.); wangxintong1999@gmail.com (X.W.); nsj8589@163.com (S.N.); 13939529735@163.com (J.H.)

**Keywords:** *Atherigona orientalis*, distribution, morphological characteristics, development duration, color preference, annual occurrence dynamics

## Abstract

*Atherigona orientalis* (Schiner 1968) is a severely underestimated agricultural pest, posing a potential threat to agriculture that far exceeds expectations. However, the general public considers *A. orientalis* to be a saprophytic and sanitary pest. This study presents the occurrence and host plants of *A. orientalis* in China, and further reports the morphological characteristics, developmental stages, and color preferences of *A. orientalis*. In addition, the population dynamics of *A. orientalis* in Changsha City have been recorded to strengthen the attention and research of researchers on this pest.

## 1. Introduction

The current situation and elaborate biological characteristics are the basis for understanding the behavior, value, and risk assessment of organisms, especially pests and invasive species [1,2,3,4]. In general, the habitat and survival of a pest can be regulated and limited by its biological characteristics, providing more countermeasures for population control and diffuse limitation of pests or invasive species [5,6]. For instance, the attractants currently in use to control pests have been developed primarily based on their feeding and reproduction characteristics [7,8,9]. Sex pheromones or food lures developed based on the biological characteristics of agricultural pests, such as *Bactrocera cucurbitae* (Coquillet 1899) (Diptera: Tephritidae) and *B. dorsalis* (Hendel 1984) (Diptera: Tephritidae), have demonstrated strong green control capabilities in controlling population sizes [10,11,12]. For *B. mianx* (Enderlein 1898) (Diptera: Tephritidae), for which specific lures have not been developed yet, control could be mainly achieved through the use of a green sticky ball (invented based on the preferred color of *B. mianx*) [13]. Controlling several key biological characteristics can effectively limit the colonization and spread of pests [14,15,16].

*Atherigona orientalis* Schiner (1868) is an “underrated” invasive pest able to infest over 50 varieties of fruits and vegetables in 26 families (data from: https://www.cabi.org/). However, attention to this pest has focused primarily on its carrion-feeding characteristics and not its phytophagous activities [17,18,19]. In effect, the damage to agriculture caused by *A. orientalis* is more serious than the public considers. Generally, the adult individuals lay the eggs on the cracks or grooves of the hosts, such as the permanent calyx of pepper *Capsicum annuum* (Linnaeus 1753) (Solanales: Solanaceae), further feeding on the flesh of the host by the hatchable larvae [20]. In addition, *A. orientalis* is able to develop in decaying fruit as well as animal carcasses or feces, resulting in the proliferation of a significant quantity of germs [21] which would accelerate decay and damage to crops in the pest-infested area. As a pantropical pest, *A. orientalis* has a predominant distribution ranging from 20° N to 20° S [22,23,24]. Countries such as South Korea, the Dominican Republic, and Cambodia have classified *A. orientalis* as a quarantine pest. The United States has included in the entire genus of *Atherigona* on its list of quarantine pests since 2017 (http://www.pestchina.com/#/) [25,26]. In China, *A. orientalis* has been documented in most southern provinces, including Zhejiang, Hunan, Fujian, Guangdong, and Guangxi, as well as in certain northern provinces such as Hebei and Ningxia, where its presence has been reported in association with decaying fruits [25,27]. Despite the fact that *A. orientalis* are present in many regions across the globe, the precise location of the origin and how they disseminated remains to be elucidated. With the climate warming and increased trade, there is a risk of *A. orientalis* spreading northwards, even though no colonization of *A. orientalis* has been found in Northern regions yet [18,19,28]. Additionally, due to the harsh survival habitat, large-scale outbreaks of *A. orientalis* populations with pathogens are also prone to ecological degradation and biodiversity loss. To date, unfortunately, data on the biological characteristics of *A. orientalis* are rarely reported, which is essential to supply the related information to develop effective countermeasures.

In the current study, we investigated the distribution of *A. orientalis* in Hunan Province (China), described the key morphological characteristics of adult *A. orientalis* that can be used to distinguish them from other closely related species, and further described the basic morphological characteristics of *A. orientalis* in different immature stages (including eggs, 3-instar larva, and puparium). In addition, the development duration, host plant, different color tendencies, and population dynamics of *A. orientalis* were explored to provide valuable information on the biological characteristics for a subsequent prevention and control strategy.

## 2. Materials and Methods

### 2.1. Distribution and Hosts Survey of the Atherigona orientalis in Hunan Province

Ranging from June 2020 to December 2022, a total of 41 sites in 14 prefectural and municipal cities in Hunan Province were sampled for 20 rotting melons and fruits in total, including chili pepper *Capsicum annuum* (Linnaeus 1753) (Solanales: Solanaceae), bitter melon *Momordica charantia* (Linnaeus 1753) (Cucurbitales: Cucurbitaceae), loofa *Luffa aegyptiaca* (Miller 1768) (Cucurbitales: Cucurbitaceae), tomato *Solanum lycopersicum* (Linnaeus 1753) (Solanales: Solanaceae), eggplant *Solanum melongena* (Linnaeus 1753) (Solanales: Solanaceae), muskmelon *Cucumis melo* (Linnaeus 1753) (Cucurbitales: Cucurbitaceae), yellow peach *Prunus persica* (Linnaeus 1753) (Rosales: Rosaceae), and others (Appendix A). The longitude, latitude, and collection date are listed in Appendix A. Decaying crops collected as described above were further cultivated in the insect cages (30 cm × 30 cm × 30 cm) in an artificial climate incubator with the conditions of temperature 25–30 °C, humidness 70 to 80%, photoperiod 14 L:10 D. The insect cages were checked daily to determine if any flies had emerged, and the adults were identified based on morphological characteristics. The emerged adults were collected according to the sampling region and different hosts.

### 2.2. Morphological Characterization of Different Developmental Stages of Atherigona orientalis

The morphology of *A. orientalis* at different developmental stages (including the adult, egg, larva, and pupa) was observed and photographed using the stereoscope (Leica S8AP0, Leica, Wetzlar, Germany). The photographs were synthesized using a Helicon Focus 6 system. The male and female adults were presented with the habitus in lateral view, thorax in dorsal view, head in lateral and overhead view, legs and abdomen in dorsal view, and further marked the main differences in legs and abdomen between males and females. The eggs were principally characterized by their morphology and surface reticulation. The larva was chiefly described in terms of its habitus in lateral view. The pupa was described in terms of the color change in the puparium and ring folds of the posterior peritremes.

### 2.3. Developmental Duration of Each Insect State of Atherigona orientalis

The infested crops with the larvae of *A. orientalis* were collected from Wangcheng vegetable land (Changsha, Hunan: 112.80° N, 28.37° E), further rearing on fresh peppers *C. annuum* with experimental conditions of photoperiod 14 L:10 D, temperature 28 ± 1 °C, and relative humidness 60 to 80%, until they pupate naturally to larvae. The feathered adults were reared separately with yeast, brown sugar, and distilled water at 1:3:50.

Five pairs of adult *A. orientalis* were placed in each cage and repeated eight times (40 pairs of *A. orientalis* in total). The eggs were collected from chili peppers twice daily at 12:00 a.m. and 8:00 p.m. using a water-dipped brush. The eggs were then counted and preserved on moist filter paper in a Petri dish. Daily observations were processed to record the hatching of eggs, the pupariation of larvae, the eclosion times, and the time of longevity of the adults. In this part, a total of 40 adult pairs, 80 eggs, 60 larvae, and 50 pupae were observed and recorded.

The Weibull distribution model was used to anticipate the expected longevity of male and female adult *A. orientalis*. The cumulative mortality curve of adults, scale parameter (η), and shape parameter (β) were produced and visualized using Matlab software (v R2024b). LT50 values (the time interval after which the population of the parasitoid was reduced to half) were also calculated from fitted Weibull curves.

### 2.4. The Color Selections of Adult Atherigona orientalis

The transparent shellac was evenly spread on the different color (including red, orange, yellow, green, blue, purple, and black) cardstocks (7.5 × 7.5 cm). The glued cardstocks were randomly pasted on a well-lit wall, and the number of adult *A. orientalis* caught on different color cardstocks was counted at 1-month intervals for a period of three months (30 September 2021–30 December 2021). Duncan’s multiple range test (SPSS v 13.0) was used to analyze the significant difference between each host and cardstock (*p* < 0.05).

### 2.5. Monitoring of Atherigona orientalis Population Dynamics

The population dynamics of *A. orientalis* were monitored in fruit and vegetable fields around the campus of Hunan Normal University in Yuelu District (Changsha, Hunan). Based on field visits, 7 fruit and vegetable plots were selected as a test site for the survey, distributed at the junction of residential areas and the Yuelu Mountain forest, with crop shops, supermarkets, open dumps, and canteens as well (Appendix A). The information on test sites is listed in Table 1. Owing to the previous study on the color selection, yellow cardstocks were shellac-dipped and applied as a monitoring tool, using 5 cardstocks for each site. The yellow cardstocks were required to hang on a pergola or branch about 1.5 m above the ground, not shaded by leaves or in direct sunlight.

The research was conducted from 1 March 2022 to 31 March 2023, of which no recording was conducted from 1 December 2022 to 15 February 2023 owing to the COVID-19 pandemic. During the low peak period of *A. orientalis* population, 1 March 2022 to 1 June 2022 and 1 October 2022 to 31 March 2023, the cardstocks were changed once a month, while in the peak population period, 1 June 2022 to 1 October 2022, the cardstocks were replaced every 10 days. The number of *A. orientalis* on replaced cardstocks was counted, and they were stored in 75% ethyl alcohol.

## 3. Result

### 3.1. Distribution and Host Ranges of Atherigona orientalis in Hunan Province

Except for cucumber, plum, peach, fig, and red dates, *A. orientalis* was found in all 15 investigated host plant species (Table 1). In addition to *A. orientalis*, several Tephritidae species emerged from hosts as well, including *Bactrocera cucurbitae* (Coquillet 1899) (Diptera: Tephritidae), *Bactrocera latifrons* (Hendel 1984) (Diptera: Tephritidae), *Bactrocera dorsalis* (Hendel 1984) (Diptera: Tephritidae), *Bactrocera minax* (Enderlein 1898) (Diptera: Tephritidae), and *Bactrocera ruiliensis* (Wang 2008) (Diptera: Tephritidae) (Table 1). In terms of the number of emerging *A. orientalis*, the largest number of 1392 was from chili pepper, followed by bitter melon (694), luffa (163), tomato (117), eggplant (72), melon (61), yellow peach (44), and tangerine (42). Further combined with the hatching rate, several hosts with large emergence numbers had high percentages as well, such as chili pepper (99.43%), tomato (100%), bitter melon (84.62%), and eggplant (100%) (Table 1). On the whole, *A. orientalis* was found in most investigated hosts with large numbers, of which the chili peppers were infested most by *A. orientalis*.

### 3.2. Morphological Characteristics of Atherigona orientalis in Various Stages

The morphological characteristics of *A. orientalis* in different developmental stages, including adults, eggs, larvae, and puparium, were described. The adults were mainly described in terms of the key morphological characteristics to differentiate them from other closely related species. The detailed morphological descriptions of adult *A. orientalis* can be referred to in Suh SJ and Pont [17,29]. The descriptions of *A. orientalis* eggs, larvae, and puparium were based to Couri [30], and Grzywacz [26], and Ferrar [31], separately.

#### Key Morphological Characteristics of *Atherigona orientalis* Adults

The side views of female and male individuals are listed in Figure 1A,B, with the female body length ranging from 4.7 to 5.2 mm and the male body length ranging from 3.2 to 4.3 mm (Figure 1A,B). The following key biological characteristics distinguish *A. orientalis* from other closely related species: 1: the basal lateral setula of scutellum at least one-third as long as sub-basal lateral seta (Figure 1C,D); 2: the male palpi are elongate but not clubbed, with fine hairs along the apical ventral part (Figure 1E); 3: the *r-m* beyond middle of cell dm (Figure 1F); 4: the male fore femur displayed a shallow dorsal preapical excavation with dense setulae (Figure 1G); 5: trifoliate process absent (Figure 1H).

**Eggs:** The eggs of *A. orientalis* are cylindrical-like, about 0.82 in length and 0.2 mm in width, and appear creamy-white with a polygonal reticulated structure on the surface. The reticulation is formed by 16–18 longitudinal ribs intersecting fine transverse striae. Moreover, the extremities have narrow, centrolateral projections (Figure 2A).

**Larvae:** We summarized the basal characteristics of 3 days-old *A. orientalis* larva for preliminary identification. The body length of 3 days-old larva ranges from 4.0 to 5.6 mm with 12 typical calyptrate body segments in total, including 1 pseudocephalon (upper left of Figure 2B), 3 thoracic segments, 7 abdominal segments, and 1 anal segment carrying posterior spiracles (Figure 2B). The black poststigmas of 3 days-old larva are distinctly raised with 3 curvilinear ostiums (bottom right of Figure 2B).

**Puparium:** The puparium of *A. orientalis* is barrel-shaped, with yellow or orange color in the early stages of pupation. The color of the puparium deepens to a reddish brown over time. Width 1.04–1.18 mm and length 3.42–3.85 mm (Figure 2C–E). Posterior spiracles with three straight slits and anterior spiracles conical were bearing 6–8 lobes (Figure 2F,G).

### 3.3. The Development Duration of Atherigona orientalis in Different Stages

Under the conditions of photoperiod 14 L:10 D, temperature 28 ± 1 °C, and relative humidity of 60 to 80%, the developmental durations of the egg, larva, and pupa of the *A. orientalis* were 2 to 3, 5 to 7, and 6 to 8 days, respectively. Moreover, without nutritional supplementation, females could survive for 2–3 days and males for 1–2 days. In contrast, when fed normally, females survived for a maximum of 15 days with an average of 11 days, while males for 12 days and 8 days on average, respectively (Table 2).

The cumulative mortality curves of male and female *A. orientalis* adults are shown in Figure 3. The proportion surviving at each time interval is presented along with the fitted Weibull curve, and the observed longevity basically overlaps with the predicted curves (Figure 3). The values of Weibull parameters (η and β) and LT50 are included in Table 3.

### 3.4. The Color Selections of Adult Atherigona orientalis

Figure 4 shows the results of color selection by *A. orientalis*. The significant differences in the color selection by *A. orientalis* were analyzed using Duncan’s multiple range test (SPSS v 13.0). There is a significant difference in seven colors, of which the percentage of yellow and green is significantly higher than other colors (*p* < 0.05) (Figure 4 and Appendix A). For the first and second month, the selection followed with yellow and green > red, orange, blue, purple, and black. For third month, the selection followed with green and yellow > orange > red, blue, purple, and black. In general, of the seven colors an adult *A. orientalis* prefers yellow and green.

### 3.5. Dynamics Monitoring of Atherigona orientalis Around Changsha City

According to the results of color selections, yellow sticky traps were used to monitor changes in population dynamics of *A. orientalis*. From 1 March 2022 to 31 March 2023, the population dynamics of *A. orientalis* were monitored for nearly one year at the seven sites around Changsha City. The results have shown that *A. orientalis* occurred from early March, and the population rose gradually with increasing temperature till to June. After June, the *A. orientalis* population increased rapidly owing to the scorching temperatures until September, when the population began to decline quickly as the temperature cooled. Figure 4 has shown that the changes in population were strongly influenced by local temperature. Additionally, the caught number of male and female individuals suggests that the sex composition of *A. orientalis* was more females and fewer males (Figure 5 and Appendix A).

## 4. Discussion

Host range, as one of the reference factors for pest damage risk assessment, is a main limitation for the dispersal of pests as well and a wider host range for a pest is more difficult to control and manage [32,33,34]. Correspondingly, varied host ranges would further promote the rapid spread or invasion of pests [35,36]. *A. orientalis* is both a phytophagous and saprophagous insect with a diverse host range and staggering adaptability, resulting in its widespread distribution in the pantropic. Furthermore, its initial origin is unclear [17,37]. In China, the documented spread of *A. orientalis* encompasses all the southern provinces and several northern provinces (http://col.especies.cn/). In this study we collected 20 usual carrion crops in 14 prefectures and municipalities in Hunan Province, to investigate the distributions and hosts of *A. orientalis*, of which the chili pepper was infested most seriously. Our results were consistent with those of Ogbalu et al., who found that *A. orientalis* infested a large number of crops, mainly chili peppers and tomatoes [38]. Hunan is one of the provinces with the largest area of chili pepper cultivation in China, which is generally sown in April and harvested from July to September [39], which is the peak of *A. orientalis* population. The ripening of chili peppers provided a hospitable environment for the *A. orientalis*, which in turn led to an increase in population. The massive infestation in turn led to detached rotting of the chili peppers which further led to an increase in the population of *A. orientalis* owing to the saprophytic characteristics, and ultimately this led to a negative cycle [23,25]. Therefore, the control measures for adult *A. orientalis* should be processed before June, which is before the fruiting period of chili peppers.

The initial step in comprehending insects is the mastery of their morphological characteristics, especially the pests. Currently, the utilization of visual representations to expedite the identification and recognition of pests is emerging as a formidable tool for initial management strategies [40,41]. For *A. orientalis* adults, many studies have provided the key morphological characteristics to distinguish closely related species of the same genus [17,19,28]. Bohart and Greesitt [42] created the cephalopharyngeal skeleton of the 1 to 3 days-old larvae of *A. orientalis* and the overall schematic diagram of the 3rd instar larva. Couri and AraúJo [30] briefly characterized the three larval instars and provided the full view cephalopharyngeal skeleton, prostigma and poststigma, and ventral creeping welt diagrams for each instar larva. In 2014, Grzywacz and Pape [43] reported the ink illustrations and color plates of skull structures of the 1st to 3rd *A. orientalis* larva, and further described the morphology of larva from the 1st to 3rd instar using optical and scanning electron microscopy. Concerning the morphological characteristics of pupae, there are fewer descriptions of *A. orientalis* pupae, with only Skidmore [22] briefly describing the cephalopharyngeal skeleton of the pupa. In this study, we characterize the key morphological characteristics of *A. orientalis* adults as well as the basic morphological characteristics of other immature stages, intending to provide usable data for agricultural and ecological protections at the first opportunity.

Previous researchers tested the developmental duration of *A. orientalis* under different conditions. Skidmore et al. [22] (under 29.44 °C) reported that the developmental duration of eggs, larvae, and pupa needed 12 h, 5 d, and 6 d, respectively. Herawani et al. [20] (under the temperature 25–27 °C and relative humidity 60%) reported that the eggs, larvae, and pupa needed 1.62 d, 11.93 d, and 5.08 d, respectively, and that the average longevity of females was 32.85 d and males was 31.40 d. In this study, the developmental duration of *A. orientalis* eggs, larvae, and pupa needed 2–3 d, 5–7 d, and 6–8 d, respectively, under the conditions of photoperiod 14 L:10 D, temperature 28 ± 1 °C, and relative humidity 60–80%. Under the same conditions the average longevity of females was 3–15 d and males was 1–12 d, which indicates that the developmental durations of *A. orientalis* were affected by different environmental conditions and so the control strategies need to be flexible according to the local environmental factors.

For color selection, the *A. orientalis* numbers on yellow and green cardstocks were significantly higher than other colors, suggesting that the preference for color by *A. orientalis* might be related to the preferred hosts, such as Solanaceae and Cucurbitaceae crops. Furthermore, creating or developing efficient control strategies and products based on the color preferences of pests is also one of the important ways to control pests. For instance, *Bactrocera minax* and *B. dorsalis*, as two destructive pests, were found to have a preference for green and yellow colors, respectively, for which researchers have developed color-specific traps with remarkable success [44,45,46]. In the current study, *A. orientalis* has a distinct color preference as well, which could provide data for the development of control products for this pest.

Various factors can influence the population dynamics of pests, of which the climate and host changes take a crucial role in pest management [47,48,49]. In the current study, the population of *A. orientalis* around the Yuelu Mountain was strongly linked to changes in local climate and host plants and the population peak ranged from June to September. As a pantropical euryphagy pest, *A. orientalis* is sensitive to changes in climate and can feed on crops over 26 families [37,38]. Locally, the temperature usually rapidly increases in June and gradually declines after September, while the Solanaceae crops such as chili peppers are generally sown in April and harvested in September. Furthermore, rainfall in Hunan Province is concentrated during the summer months, resulting in a hot and humid environment that is conducive to the rapid deterioration of crops, potentially providing more suitable living space for both *A. orientalis* adults and larvae [50,51,52]. Food scraps or garbage generated by humans may also deteriorate under conditions of high temperature and humidity, thereby becoming a potential food source or host for *A. orientalis*. The above conditions might provide a suitable living environment for *A. orientalis*. Therefore, according to the biological characteristics we found that more targeted management strategies need to be implemented before the rapid population growth, especially before June. Meanwhile, the culling after September also needs to be phased in to minimize the population of *A. orientalis* the following year.

## 5. Conclusions

In summary, *A. orientalis* has been found throughout Hunan Province, owing to the “unnoticed” characteristics and powerful adaptability in the hosts and climate. Among the various infested hosts, chili peppers and tomatoes were damaged mostly. The development durations, color preferences, and morphological characteristics of *A. orientalis* in different stages have been described in detail, which can be used as reference data for the methods of controlling *A. orientalis*. Except for a few extremely cold months, *A. orientalis* occurs almost throughout the year, meaning that *A. orientalis* has extremely strong adaptability to the environment and can survive for generations under unfavorable conditions. This study provides necessary information about *A. orientalis* regarding its biological characteristics and Hunan distribution, with the aim of giving more attention to this pest and providing data supplements for subsequent management strategies.

## Figures and Tables

**Figure 1 insects-16-00931-f001:**
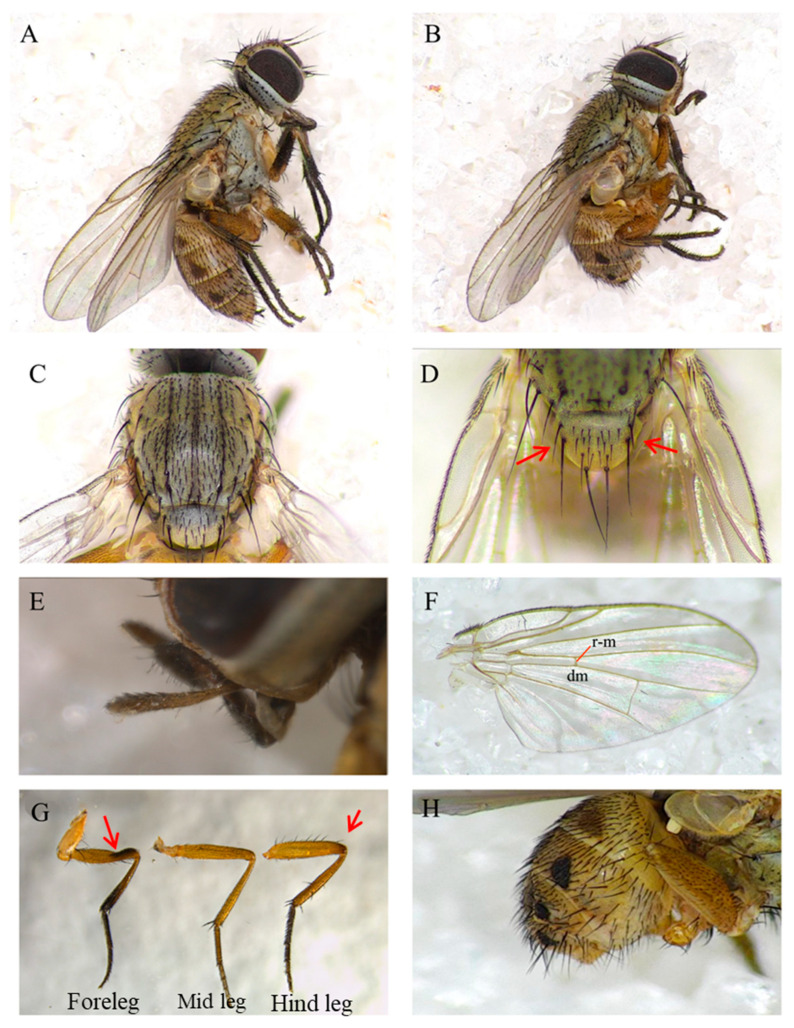
Key morphological characteristics of *Atherigona orientalis* (Schiner 1968) (Diptera: Muscidae) adults: (**A**) side view of female *A. therigona*; (**B**) side view of male *A. orientalis*; (**C**) thorax; (**D**) basal pair of scutellarsetae; (**E**) palpi in male; (**F**) wing, dm: discal medial cell, and r-m: radial-medial crossvein; (**G**) the foreleg, mid leg, and hind leg of males; and (**H**) male terminalia.

**Figure 2 insects-16-00931-f002:**
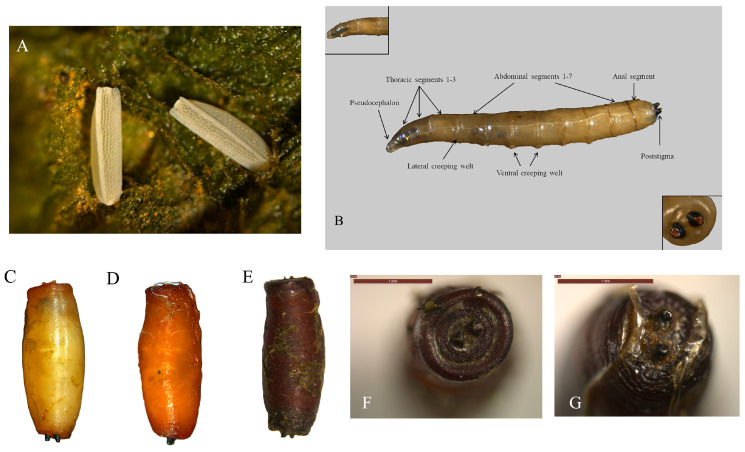
Morphologies of eggs (**A**), larva (**B**), and puparium (**C**–**G**) of *Atherigona orientalis* (Schiner 1968) (Diptera: Muscidae).

**Figure 3 insects-16-00931-f003:**
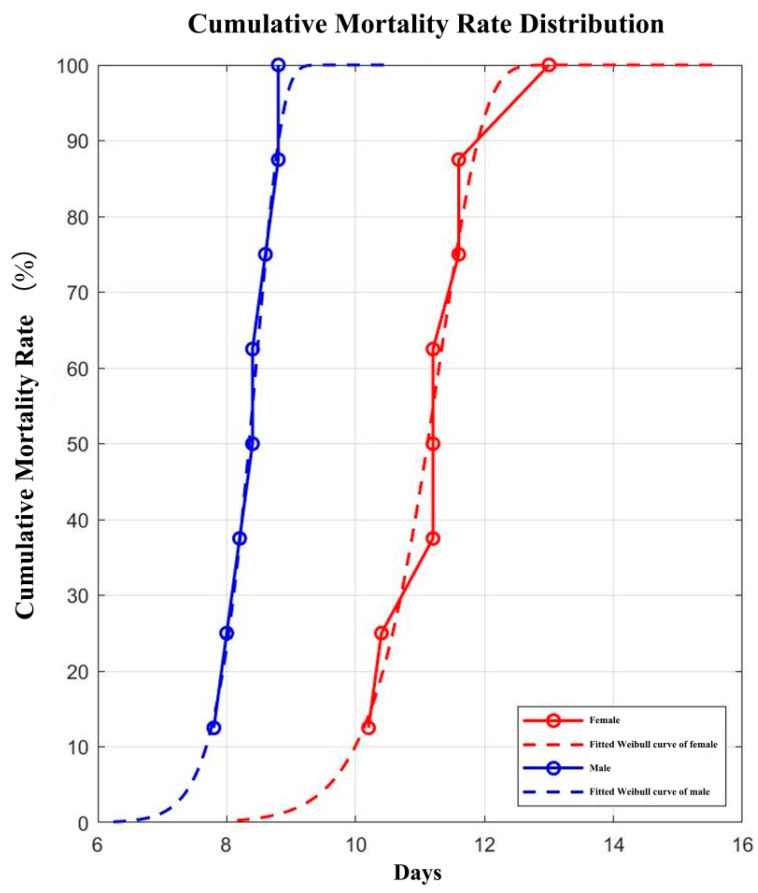
Cumulative mortality curves of male and female *Atherigona orientalis* adults.

**Figure 4 insects-16-00931-f004:**
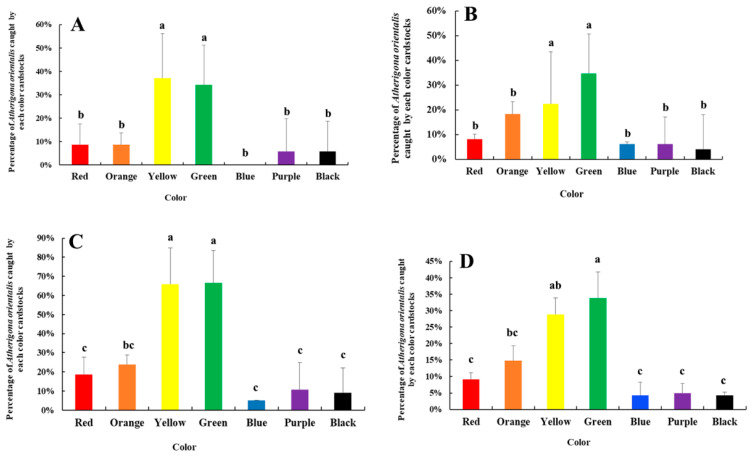
The selection of different colored cardstocks by *Atherigona orientalis* (Schiner 1968) (Diptera: Muscidae). (**A**) First month; (**B**) second month; (**C**) third month; (**D**) total. The different letters above the bars represent the significant difference (*p* < 0.05).

**Figure 5 insects-16-00931-f005:**
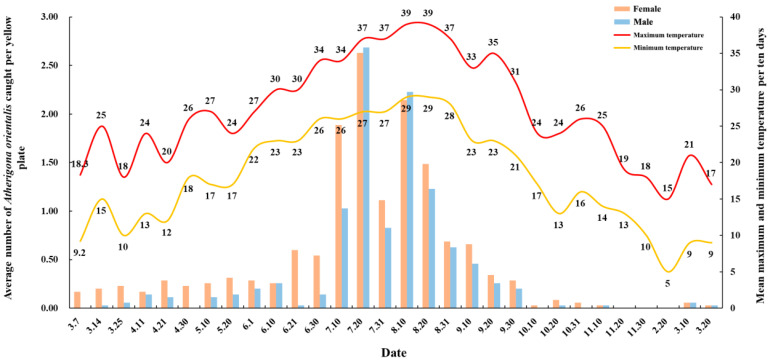
The population dynamics of the *Atherigona orientalis* (Schiner 1968) (Diptera: Muscidae) in the surrounding areas of Changsha City (Survey results every 10 days) (March 2022−March 2023).

**Table 1 insects-16-00931-t001:** Occurrence of *Atherigona orientalis* (Schiner 1968) (Diptera: Muscidae) in Hunan Province.

Family	Host Plants	Percentage of Flies Emerging from Host Plants				
		*Atherigona orientalis*	*Bactrocera tau*	*Bactrocera latifrons*	*Bactrocera dorsalis*	*Bactrocera minax*
Solanaceae	*Capsicum annuum*	1392 (99.43%)	3 (0.20%)	5 (0.37%)	-	-
	*Lycopersicon esculentum*	117 (100%)	-	-	-	-
	*Solanum melongena*	72 (100%)	-	-	-	-
Cucurbitaceae	*Momordica charantia*	694 (77.50%)	201 (22.40%)	-	1 (0.10%)	-
	*Luffa cylindrica*	163 (57.00%)	123 (43.00%)	-	-	-
	*Cucumis melo*	61 (76.25%)	19 (23.75%)	-	-	-
	*Cucurbita pepo*	11 (84.62%)	2 (15.38%)	-	-	-
	*Benincasa hispida*	11 (100%)	-	-	-	-
	*Cucurbita moschata*	6 (2.64%)	221 (97.36%)	-	-	-
	*Lagenaria sciceraria*	3 (100%)	-	-	-	-
	*Cucumis sativus*	-	-	-	-	-
Rosaceae	*Amygdalus persica*	44 (100%)	-	-	-	-
	*Prunus salicina*	-	-	-	-	-
	*Prunus persica*	-	-	-	-	-
	*Pyrus* spp.	1 (100%)	-	-	-	-
Rutaceae	*Citrus reticulata*	42 (91.30%)	-	-	1 (2.20%)	3 (6.50%)
Actinidiaceae	*Actinidia chinensis*	1 (100%)	-	-	-	-
Ebenaceae	*Diospyros kaki*	3 (100%)	-	-	-	-
Moraceae	*Ficus carica*	-	-	-	-	-
Rhamnaceae	*Ziziphus jujuba*	-	-	-	3 (100%)	-

Note: The numbers on the left of brackets represent the eclosion quantity of each type of fly, and the data in brackets is the percentage of the eclosion quantity of each type of fly in the total eclosion quantity of the corresponding host plant; “-” represents the absence of eclosion in the host plan.

**Table 2 insects-16-00931-t002:** The development duration of *Atherigona orientalis* (Schiner 1968) (Diptera: Muscidae) in different states.

Stage	Number	Development Time (Days)	Mean Developmental Time (Days)
Egg	80	2–3	2.19 ± 0.25
Larva	60	5–7	5.56 ± 0.49
Pupa	50	6–8	6.16 ± 0.38
Female adult	40	3–15	11.3 ± 2.43
Male adult	40	1–12	8.38 ± 1.98

**Table 3 insects-16-00931-t003:** Weibull parameter values for cumulative mortality of *Atherigona orientalis*.

	Male	Female
η	8.4856	11.3475
β	22.5861	17.6779
LT50W	8.3490	11.1146
LT50O	8.400	11.200

## Data Availability

The original contributions presented in this study are included in the article and Appendix A. Further inquiries can be directed to the corresponding author.

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
