# Peer review of "Occurrence, Biological Characteristics, and Annual Dynamics of Atherigona orientalis (Schiner 1968) (Diptera: Muscidae) in China"

_insects, 2025, doi:10.3390/insects16090931_

Round 1
Reviewer 1 Report
Comments and Suggestions for Authors
- Title: Insert year and author.
- Simple Summary: Atherigona orientalis: Insert author and year.
- Please do not remove words from the title to compose the keywords.
- Capsicum annuum: Insert author and year, family, and order.
- chili pepper Capsicum annuum, bitter melon Momordica charantia, loofa Luffa aegyptiaca, tomato Solanum lycopersicum, eggplant Solanum melongena, muskmelon Cucumis melo, yellow peach Prunus persica et al.: Insert the author, year, family, and order in the scientific names.
- Touch: The hatched adults are “ by “the emerged adults.
- Bactrocera cucurbitae, B. latifrons, B. dorsalis, B. minax, and ruiliensis: Enter names in full, author, year, author, family, and order.
- Change “hatched” to “emergency”.
- Figure 1. Side views of orientalis: Insert author and year, family, and order (The figures are self-explanatory).
- Table 2. The development duration of orientalis: Insert author and year, family, and order (The figures are self-explanatory).
- In the results and discussion, the authors used data collected from species of the Tephritidae family; however, this is not mentioned in the objectives or introduction of the article.
- Observe the International Standards of Zoological and Botanical Nomenclature.
Author Response
Dear Editor and Reviewers:
Thank you very much for providing us with this great opportunity to submit our revised manuscript entitled “Occurrence, Biological characteristics, and Annual occurrence dynamics of Atherigona orientalis (Schiner 1968) (Diptera: Muscidae) in China”. We deeply appreciate the time and effort that you’ve spent reviewing our manuscript. Those detailed and constructive comments are all valuable to improve our manuscript. Consequently, we checked and revised the points mentioned in the comments. Please see the tracked version. The main corrections in the paper, along with the point-by-point responses to the reviewers’ comments, are as follows. To facilitate the discussion, we first retype your comments, which are colored in blue, and then present our responses to them.
Comments 1: Title: Insert year and author.
Response: Thank you for the valuable suggestion. We have inserted the year and author behind Atherigona orientalis.
Comments 2: Simple Summary: Atherigona orientalis: Insert author and year.
Response: Thank you for the valuable suggestion. We have inserted the year and author behind Atherigona orientalis.
Comments 3: Please do not remove words from the title to compose the keywords.
Response: Thank you for your suggestion. We have revised the title of the paper to make it more relevant to the content of this study.
Comments 4: Capsicum annuum: Insert author and year, family, and order.
Response: Thank you for the valuable suggestion. We have added the author, year, family, and order behind Capsicum annuum.
Comments 5: chili pepper Capsicum annuum, bitter melon Momordica charantia, loofa Luffa aegyptiaca, tomato Solanum lycopersicum, eggplant Solanum melongena, muskmelon Cucumis melo, yellow peach Prunus persica et al.: Insert the author, year, family, and order in the scientific names.
Response: Thank you for the valuable suggestion. We have added the author, year, family, and order behind all the scientific names.
Comments 6: Touch: The hatched adults are “ by “the emerged adults.
Response: Thank you. We have changed the “hatched adults” to “emerged adults”.
Comments 7: Bactrocera cucurbitae, B. latifrons, B. dorsalis, B. minax, and B. ruiliensis: Enter names in full, author, year, author, family, and order.
Response: Thank you for the valuable suggestion. We have supplemented the full scientific names, added the year, author, family, and order as well.
Comments 8: Change “hatched” to “emergency”.
Response: Thank you for the valuable suggestion. We have changed all “hatched” to “emergency” in the whole manuscript.
Comments 9: Figure 1. Side views of orientalis: Insert author and year, family, and order (The figures are self-explanatory).
Response: Thank you for the valuable suggestion. We have inserted author, year, family, and order after A. orientalis. We hope to keep the figure legends so that others who are unprofessional can read it easier.
Comments 10: Table 2. The development duration of orientalis: Insert author and year, family, and order (The figures are self-explanatory).
Response: Thank you for the valuable suggestion. We have inserted author, year, family, and order after A. orientalis.
Comments 11: In the results and discussion, the authors used data collected from species of the Tephritidae family; however, this is not mentioned in the objectives or introduction of the article.
Response: Thank you for the valuable suggestion. We have supplemented the information about the Tephritidae family in the first paragraph of the introduction.
Comments 12: Observe the International Standards of Zoological and Botanical Nomenclature.
Response: Thank you for the valuable suggestion. We have gained a thorough understanding of the International Standards of Zoological and Botanical Nomenclature. It will greatly enhance the publication or reporting of our subsequent research.
Once again, thank you for all your suggestions on this paper and for your time and effort.

Reviewer 2 Report
Comments and Suggestions for Authors
Supplementary materials were not available.
The authors didn’t inform us where they took the terminology used to describe the stages of A. orientalis. This is important, since the terminology used differs from that used in other studies.
The statistics used for biology studies were elementary. No Weibull Distribution was made to compare the expected and the observed longevity of the adults. Also, the entire developmental experiment needs to be reanalyzed. No viability/mortality for each period was provided. Did the authors observe those specimens in a single repetition? For this kind of experiment, a minimum of 3 repetitions is required.
The authors wrote that the description of A. orientalis focused on their immature stages and “described” the adult stage in this study; however, Schiner, in 1868, did the first description of the species. Did the authors try to find this paper? Similarly, the morphological characteristics provided in this manuscript do not allow us to identify the species; the description made was only superficial.
The authors didn’t mention in the methodology the statistical analysis they used for the entire study.
In the discussion, the authors say that the population studied in the paper was strongly linked to changes in local climate and host plants. However, they didn’t mention the temperature for the period, the rain, or other climate factors.
Author Response
Dear Editor and Reviewers:
Thank you very much for providing us with this great opportunity to submit our revised manuscript entitled “Occurrence, Biological characteristics, and Annual occurrence dynamics of Atherigona orientalis (Schiner 1968) (Diptera: Muscidae) in China”. We deeply appreciate the time and effort that you’ve spent reviewing our manuscript. Those detailed and constructive comments are all valuable to improve our manuscript. Consequently, we checked and revised the points mentioned in the comments. Please see the tracked version. The main corrections in the paper, along with the point-by-point responses to the reviewers’ comments, are as follows. To facilitate the discussion, we first retype your comments, which are colored in blue, and then present our responses to them.
Comments 1: Supplementary materials were not available.
Response: We apologize for the unavailability of the Supplementary materials. We have re-uploaded the Supplementary materials to the system.
Comments 2: The authors didn’t inform us where they took the terminology used to describe the stages of A. orientalis. This is important, since the terminology used differs from that used in other studies.
Response: We apologize for the inappropriate wording in morphological descriptions of A. orientalis. We have referenced descriptions from other studies of the morphological terminology on A. orientalis and rewritten the morphological descriptions.
Comments 3: The statistics used for biology studies were elementary. No Weibull Distribution was made to compare the expected and the observed longevity of the adults. Also, the entire developmental experiment needs to be reanalyzed. No viability/mortality for each period was provided. Did the authors observe those specimens in a single repetition? For this kind of experiment, a minimum of 3 repetitions is required.
Response: Thank you for the valuable suggestion. We have supplemented the cumulative mortality curve of male and female A. orientalis adults using the Weibull distribution model, and the key parameters (η, β, LT50W, LT50O). We apologize for not providing information on the viability and mortality of A. orientalis in different stages. We set up 8 replicates for the adult A. orientalis (5 pairs each cage, 40 pairs in total). However, the number of eggs, larvae, and pupae per cage is different, so we integrated them for statistical purposes without setting duplicates. The specific numbers of eggs (80), larvae (60), and pupae (50) we observed were all alive, and all samples were from 8 repetitions of adult A. orientalis. We did record the hatchability of eggs, the pupation rate of larvae, and the eclosion rate of pupae. However, the data mentioned above were integrated and utilized in our next study of A. orientalis, which is currently being written. We sincerely hope you could understand how we allocate and use the data, and we hope you could continue to provide guidance and review for our future research.
Comments 4: The authors wrote that the description of A. orientalis focused on their immature stages and “described” the adult stage in this study; however, Schiner, in 1868, did the first description of the species. Did the authors try to find this paper? Similarly, the morphological characteristics provided in this manuscript do not allow us to identify the species; the description made was only superficial.
Response: We apologize for the unprofessional description of A. orientalis. We have carefully reviewed the papers of other researchers and referenced the professional terminology. We have shifted the focus of our description of A. orientalis from a general morphological overview to the key morphological characteristics that distinguish it from other closely related species (Section 3.2.1 of the Results part).
Comments 5: The authors didn’t mention in the methodology the statistical analysis they used for the entire study.
Response: We apologize for the missing information on the statistical analysis. We have supplemented the statistical analysis information in both the method and result parts of the color selection of A. orientalis. For the developmental duration, we have supplemented the Weibull Distribution model to compare the expected and the observed longevity of the adult A. orientalis.
Comments 6: In the discussion, the authors say that the population studied in the paper was strongly linked to changes in local climate and host plants. However, they didn’t mention the temperature for the period, the rain, or other climate factors.
Response: Thank you for the valuable suggestion. We have supplemented the discussion about the possible effects of rainfall on changes in the A. orientalis population.
Once again, we sincerely thank you for all your suggestions on this paper. It greatly improves the professionalism and quality of our articles.
Reviewer 3 Report
Comments and Suggestions for Authors
The manuscript presents interesting and important results of a study focusing on A. orientalist biology and morphology. However, some aspects of the manuscript do not allow its acceptance in the present form. Some parts of this contribution must be corrected or modified, especially a taxonomic/morphological part of this work.
Why Authors stated that A. orientalis is an invasive species? What is supposed oroginal distribution of this species, and which regions have been invaded recently?
Please check carefully and correct generic names and species names to italics (Atherigona, A. orientalis).
Entire genus Atherigona has been included in the US on the list of pests maybe because of A. reversura invasion (Grzywacz, A., Pape, T., Hudson, W. G., & Gomez, S. (2013). Morphology of immature stages of Atherigona reversura (Diptera: Muscidae), with notes on the recent invasion of North America. Journal of Natural History, 47(15-16), 1055-1067.)
„With the climate warming and increased trade, there is a risk of the A. orientalis spreading north-
wards, even though no colonization of the A. orientalis has been found in Northern provinces yet.” An interesting information can be that Atherigona orientalis has recently been reported from France (https://doi.org/10.1111/epp.13022), what indicate its spread towards northern regions.
The following sentence must be rewritten "In China, the A. orientalis was found in most southern provinces such as Zhejiang, Hunan, Fujian, Guangdong, Guangxi et al., and even in some northern provinces (Hebei and Ningxia) had reported that the A. orientalis was found in some rotten fruits [21,22].”
„The adults of both genders” Gender refers to cultural phenomenon, in insects biological sex (male or female) should be used instead.
Please provide morphological keys used for the most recent interpretation of A. orientalis flies. Please check the following reference https://doi.org/10.1111/syen.12215
In aims and scope of the manuscript please indicate the novelty of this study. Morphological characteristic of A. Orientalis has already been provided (in details) by previous studies. However, still a significant gap is thermal requirements for development of this species.
Please provide names of species authors for "Bactrocera cucurbitae, B. latifrons, B. dor-
salis, B. minax, and B. ruiliensis "
Please provide references which have been followed for adult, egg and larval morphology terminology. For example what do you mean by „folds of the posterior peritremes”? Respiratory slits of posterior spiracles?
Also, please mention the third install larva has been described in this study.
Adult flies description is superficial, and devoid of any species-specific characters. I recommend to include characters allowing to differentiate it from other closely related species.
Figure 2 must be improved. Eggs on fig 2A are upside down. Anterior pole of each egg is directed downwards. Please remember pupa and puparium are not the same. Especially fig 2E, it presents empty puparium. Fig 2B is difficult to interpret. I recommend to increase size of this image. Figs 2F-G do not present significant morphological features.
I do not agree with statement that adult flies of A. orientalis have not been described in details. Did you check a paper of Pont and Magpayo (1995): DOI: 10.1017/S1367426900000321
In table 2, please provide units to make it self-explanatory. Development was measured in days? I do not understand column "development duration".
„Daily observations were processed to record the hatching of eggs, the pupation of larvae, the time of pupal instar…” Pupariation, not pupation. Pupation and pupariation is not the same. What do you mean by pupal instars? Is it a new term?
Gender and sex is not the same.
Some sontences are confusing.
Author Response
Dear Editor and Reviewers:
Thank you very much for providing us with this great opportunity to submit our revised manuscript entitled “Occurrence, Biological characteristics, and Annual occurrence dynamics of Atherigona orientalis (Schiner 1968) (Diptera: Muscidae) in China”. We deeply appreciate the time and effort that you’ve spent reviewing our manuscript. Those detailed and constructive comments are all valuable to improve our manuscript. Consequently, we checked and revised the points that mentioned in the comments. Please see the tracked version. The main corrections in the paper and the point-by-point responses to the reviewers’ comments are as follows. To facilitate the discussion, we first retype your comments colored in blue and then present our responses to the comments.
Comments 1: Why Authors stated that A. orientalis is an invasive species? What is supposed oroginal distribution of this species, and which regions have been invaded recently?
Response: Atherigona orientalis is a widely known pan-tropical distributed in many tropical and subtropical regions. To date, no studies have reported the origin of A. orientalis. We believe that A. orientalis is more likely to spread gradually from tropical areas to subtropical regions and even further north. At least in central and northern China, its occurrence can be regarded as an invasion. Therefore, we currently consider A. orientalis to be an invasive pest in China. Moreover, A. orientalis have been found and reported in many north countries, such as Korea, Greece, and France et al., where are not tropical countries. In this study, we have cited several reports on the first record of A. orientalis in different countries and listed them in the introduction section.
Comments 2: Entire genus Atherigona has been included in the US on the list of pests maybe because of A. reversura invasion (Grzywacz, A., Pape, T., Hudson, W. G., & Gomez, S. (2013). Morphology of immature stages of Atherigona reversura (Diptera: Muscidae), with notes on the recent invasion of North America. Journal of Natural History, 47(15-16), 1055-1067.)
Response: Thank you for the valuable explanation. We have added the reference in our study to explain the basis for the United States listing the entire genus of Atherigona as quarantine pests.
Comments 3: With the climate warming and increased trade, there is a risk of the A. orientalis spreading northwards, even though no colonization of the A. orientalis has been found in Northern provinces yet.” An interesting information can be that Atherigona orientalis has recently been reported from France (https://doi.org/10.1111/epp.13022), what indicate its spread towards northern regions.
Response: Thank you for your valuable recommended reference. We have added this paper as a supportive reference in the Introduction part.
Comments 4: The following sentence must be rewritten "In China, the A. orientalis was found in most southern provinces such as Zhejiang, Hunan, Fujian, Guangdong, Guangxi et al., and even in some northern provinces (Hebei and Ningxia) had reported that the A. orientalis was found in some rotten fruits [21,22].
Response: We apologize for the unprofessional writing sentences. We have rewritten the sentence.
Comments 5: The adults of both genders” Gender refers to cultural phenomenon, in insects biological sex (male or female) should be used instead.
Response: Thank you for the valuable suggestion. We have changed all “genders” to male and female in our study.
Comments 6: Please provide morphological keys used for the most recent interpretation of A. orientalis flies. Please check the following reference https://doi.org/10.1111/syen.12215
Response: We apologize for the inappropriate wording in morphological descriptions of A. orientalis. We have referenced descriptions from other studies of the morphological terminology on A. orientalis and rewritten the morphological descriptions.
Comments 7: In aims and scope of the manuscript please indicate the novelty of this study. Morphological characteristic of A. orientalis has already been provided (in details) by previous studies. However, still a significant gap is thermal requirements for development of this species.
Response: Thank you for the valuable suggestion. We have shifted the focus of our description of A. orientalis from a general morphological overview to the key morphological characteristics that distinguish it from other closely related species as a novelty.
Comments 8: Please provide names of species authors for "Bactrocera cucurbitae, B. latifrons, B. dorsalis, B. minax, and B. ruiliensis".
Response: Thank you for the valuable suggestion. We have supplemented the author, tine, family, and order of these pests.
Comments 9: Please provide references which have been followed for adult, egg and larval morphology terminology. For example what do you mean by „folds of the posterior peritremes”? Respiratory slits of posterior spiracles? Also, please mention the third install larva has been described in this study.
Response: We apologize for the inappropriate wording in morphological descriptions. We carefully reviewed the morphological terminology used in other studies for different developmental stages of A. orientalis and made corrections in our study. We cited the references we had consulted in the section 3.2 as well.
Comments 10: Adult flies description is superficial, and devoid of any species-specific characters. I recommend to include characters allowing to differentiate it from other closely related species.
Response: Thank you for the valuable suggestion. We have shifted the focus of our description of A. orientalis from a general morphological overview to the key morphological characteristics that distinguish it from other closely related species.
Comments 11: Figure 2 must be improved. Eggs on fig 2A are upside down. Anterior pole of each egg is directed downwards. Please remember pupa and puparium are not the same. Especially fig 2E, it presents empty puparium. Fig 2B is difficult to interpret. I recommend to increase size of this image. Figs 2F-G do not present significant morphological features.
Response: Thank you for the valuable suggestion. We rearranged the Figure 2 and rotated the images of the eggs. The pupa has been changed to puparium in both Figure legend and main text. The Figure 2B has been enlarged. We still wanted to present the complete puparium of A. orientalis even though no significant morphological features have been presented. Therefore, we kept the image but modified the accompanying description.
Comments 12: I do not agree with statement that adult flies of A. orientalis have not been described in details. Did you check a paper of Pont and Magpayo (1995): DOI: 10.1017/S1367426900000321.
Response: We apologize for the inappropriate wording. We have checked several studies of A. orientalis decriptions, and have deleted this sentence.
Comments 13: In table 2, please provide units to make it self-explanatory. Development was measured in days? I do not understand column "development duration".
Response: Thank you for the valuable suggestion. We have added the units in the Table 2, and changed the “developmental duration” to “developmental time”.
Comments 14: Daily observations were processed to record the hatching of eggs, the pupation of larvae, the time of pupal instar…” Pupariation, not pupation. Pupation and pupariation is not the same. What do you mean by pupal instars? Is it a new term?
Response: We apologize for the inappropriate wording. We have changed “pupation” to “Pupariation” and changed the “pupal instars” to “eclosion time”.
Once again, we sincerely thank you for all your suggestions on this paper. It greatly improves the professionalism and quality of our articles.
Round 2
Reviewer 2 Report
Comments and Suggestions for Authors
Nice work on your paper.
Author Response
Comment 1:Nice work on your paper.
Response: Thank you very much for your affirmation! Your suggestions on our study are of great help. We hope to have the opportunity to cooperate with you again next time!
Reviewer 3 Report
Comments and Suggestions for Authors
I would like to thank the authors for their careful revision and for taking all comments into consideration.
The manuscript can be accepted after some minor editorial corrections:
If "et al." is used in the text, it should always appear as "et al." (with a period). However, I recommend using English phrases such as "and others" where appropriate.
The sentence "The emergencyadults were collected according to the sampling region and different hosts." should be corrected to: "The emerged adults were collected according to the sampling region and different hosts."
In Table 1, change "Atherigona oriemtalis" to "Atherigona orientalis".
In the sentence "The adults were mainly described in terms of THE key morphological characteristics to differentiate them from other closely related species," change "THE" to "the."
The sentence "The descriptions of A. orientalis eggs, larvae, and puparium were referred to Couri [31], and Grzywacz [26], and Ferrar [32], separately ." should be corrected to: "The descriptions of A. orientalis eggs, larvae, and puparium were based on Couri [31], Grzywacz [26], and Ferrar [32], respectively." Delete the space before the full stop.
Please check whether Figure 5 is correctly labeled as Figure 5, and not Figure 45.
Author Response
Dear Reviewer:
Thank you very much for correcting the careless mistakes in our study. We have made comprehensive revisions based on your feedback. The following is the response to your suggestions.
Comments 1: If "et al." is used in the text, it should always appear as "et al." (with a period). However, I recommend using English phrases such as "and others" where appropriate.
Response: Thank you for the valuable suggestion. We have appropriately changed "et al." to "the others".
Comments 2: The sentence "The emergencyadults were collected according to the sampling region and different hosts." should be corrected to: "The emerged adults were collected according to the sampling region and different hosts.
Response: We apologize for our careless negligence. We have already added a space before "adults"
Comments 3: In Table 1, change "Atherigona oriemtalis" to "Atherigona orientalis".
Response: We apologize for our careless mistake. We have changed the “Atherigona oriemtalis” to "Atherigona orientalis".
Comments 4: In the sentence "The adults were mainly described in terms of THE key morphological characteristics to differentiate them from other closely related species," change "THE" to "the.
Response: We apologize for our careless mistake. We have changed the “THE” to “the”.
Comments 5: The sentence "The descriptions of A. orientalis eggs, larvae, and puparium were referred to Couri [31], and Grzywacz [26], and Ferrar [32], separately ." should be corrected to: "The descriptions of A. orientalis eggs, larvae, and puparium were based on Couri [31], Grzywacz [26], and Ferrar [32], respectively." Delete the space before the full stop.
Response: Thank you for the valuable suggestion. We have changed the sentence correctly. We have deleted the space before the full stop as well.
Comments 6: Please check whether Figure 5 is correctly labeled as Figure 5, and not Figure 45.
Response: We have checked the title of Figure 5.